# Equity and Distributional Impact on Stunting of a Nutritional Package Targeting Children Aged 6–36 Months in China: Findings from a Modeling Study

**DOI:** 10.3390/nu12092643

**Published:** 2020-08-30

**Authors:** Zhihui Li, Wafaie W. Fawzi, Jessica L. Cohen, Stéphane Verguet

**Affiliations:** 1Department of Global Health and Population, Harvard T.H. Chan School of Public Health, Boston, MA 02115, USA; mina@hsph.harvard.edu (W.W.F.); cohenj@hsph.harvard.edu (J.L.C.); verguet@hsph.harvard.edu (S.V.); 2Vanke School of Public Health, Tsinghua University, Beijing 100084, China

**Keywords:** nutritional package, Ying Yang Bao, child stunting, equity, distributional impact, extended cost-effectiveness analysis

## Abstract

Background: Despite rapid economic development, child stunting remains a persistent problem in China. Stunting prevalence varies greatly across geographical regions and wealth groups. To address child undernutrition, the Ying Yang Bao (YYB) nutritional package has been piloted in China since 2001. Objective: We aimed to evaluate the distributional impact of a hypothetical rollout of the YYB nutritional package on child stunting across provinces and wealth groups in China, with a specific focus on equity. Methods: We used data from China Family Panel Studies and built on extended cost-effectiveness analysis methods. We estimated the distributional impact of a 12-month YYB program targeting children aged 6–36 months across 25 provinces and two wealth groups along three dimensions: the cost of the YYB program; the number of child stunting cases averted by YYB; and the cost per stunting case averted. Children in each province were divided into poverty and non-poverty groups based on the international poverty line of $5.50 per day. We also conducted a range of sensitivity analyses. Results: We showed that 75% coverage of YYB could avert 1.9 million stunting cases among children aged 6–36 months, including 1.3 million stunting cases among children living under the poverty line, at a total cost of ¥5.4–6.2 billion ($1.5–1.8 billion) depending on the type of YYB delivery. The cost per stunting case averted would greatly vary across Chinese provinces and wealth groups, ranging from ¥800 (around $220, Chongqing province) to ¥23,300 (around $6600, Jilin province). In most provinces, the cost per stunting case averted would be lower for children living under the poverty line. Conclusions: YYB could be a pro-poor nutritional intervention package that brings substantial health benefits to poor and marginalized Chinese children, but with large variations in value for money across provinces and wealth groups. This analysis points to the need for prioritization across provinces and a targeted approach for YYB rollout in China.

## 1. Introduction

As of 2015, China was home to 86 million children under 5 years of age, accounting for 13% of the world’s under-five population [1]. Despite rapid economic growth, China still has large numbers of stunted children (height-for-age Z score less than minus 2 standard deviations (SD)): it is estimated that 10% of Chinese children under five years old were stunted in 2016, equal to about 9 million children [2].

There are large disparities in child stunting rates within China, with a prevalence ranging from 4% in urban areas to 16% in rural areas, and from being almost absent in rich provinces and cities such as Beijing, Tianjin and Shanghai to being as prevalent as 29% in poor provinces and cities such as Sichuan and Gansu [2]. Previous evidence shows that stunting in children under age five could lead to impaired cognitive development, affecting educational performance and long-term economic productivity [3,4,5]. Often, children living in poor families and regions of China are more likely to be stuck in a cycle of poverty as stunting can be both a cause and a consequence of poverty [6].

To improve children’s nutritional status in China, the Ying Yang Bao (YYB) nutritional package was introduced to the country’s poor rural areas in 2001 [7]. YYB is a nutrient-dense food supplement targeting infants and young children, usually aged 6–36 months. The base of YYB is constituted of full fat soybean powder and multiple micronutrient powders, including calcium, iron, zinc and vitamins; folic acid, omega 3, omega 6, thiamin and riboflavin are also sometimes added [7,8,9]. The exact composition of YYB varies slightly, depending on the manufacturing companies. Compared to other complementary feeding supplements (e.g., sprinkles and crushable tablets) that only provide micronutrients, YYB contains soybean powder that provides both calories (usually ~50 Kcal per pack) and protein (3 g per pack) in addition to various micronutrients [7,8,9]. YYB can be directly added to children’s meals (e.g., mushy foods, soups and noodles) or reconstituted with water [10].

The YYB program was first piloted and launched in 2001 in five counties of Gansu province and targeted 4–12-month-olds [11]. Since then, it gradually expanded to other poor rural areas of China and regions affected by natural disasters. In 2011, the Chinese government invested ¥100 million ($28 million, 2011 Purchasing Power Parity (PPP)) for the YYB program covering 300,000 children in 100 counties across 10 provinces [12]. In 2017, the Chinese government issued a national nutrition strategy (for the years 2017–2030) that clearly prioritized the promotion of nutritional status of children in their first 1000 days [13]; and YYB was considered as one key intervention for rural and poor populations [13]. In 2019, the Chinese government launched the “Upgrade of YYB Plan” and has started to expand YYB coverage to all 823 counties living in poverty in the country [14].

Previous studies consistently demonstrated the effectiveness of YYB on improving children’s anthropometric status, including children’s height, in a variety of settings [15,16,17,18,19,20]. A meta-analysis summarizing publications on the effects of YYB over the time period January 2001 to March 2019 showed that YYB was associated with an average 2.46-cm (95% confidence interval (CI): 0.96–3.97) increase in children’s height and with a 40% decrease in stunting prevalence [21]. This meta-analysis covered studies from a great variety of provinces, including provinces with gross domestic product (GDP) per capita higher than $20,000 (e.g., Zhejiang and Jiangsu), provinces with moderate GDP per capita, of $10,000–20,000 (e.g., Hubei and Liaoning) and provinces with GDP per capita lower than $10,000 (e.g., Yunnan and Gansu) [21,22]. This points to the comprehensiveness of the meta-analysis and to the representativeness of the identified effect size in China. However, little work has assessed the cost-effectiveness of YYB delivery and its equity impact on child stunting across China’s provinces and wealth groups. In this study, we built on extended cost-effectiveness analysis (ECEA) methods [23,24] to study the distributional consequences of YYB rollout in China.

## 2. Methods

We developed an epidemiological model to estimate the cost, health benefits and cost-effectiveness of a hypothetical YYB program rolled out across 25 provinces and two wealth groups in China.

### 2.1. Setting

To be consistent with the previous studies evaluating the effects of YYB interventions [21], we focused on children from rural areas and aged 6–36 months, assuming that all Chinese children who satisfied the inclusion criteria would be covered by the hypothetical YYB program. The YYB intervention was assumed to be rolled out over 12 months based on current YYB practices, and the children targeted by the intervention would receive one YYB pack per day [10,25]. We examined the distributional consequences of YYB on child stunting across 25 Chinese provinces and two wealth groups. The two wealth groups were defined according to the World Bank’s international poverty line: those living under the international poverty line ($5.5 (2011 PPP) or ¥19.4 (2011 RMB) per day) [26,27] and those living above this poverty line. We simulated two coverage scenarios: a low coverage (25%) and a high coverage (75%) scenario.

### 2.2. Data Sources

We used data on stunting prevalence and household poverty status from the China Family Panel Studies (CFPS) [2]. CFPS is a nationally representative, longitudinal survey launched in 2010 and further conducted in 2012, 2014 and 2016. It covers about 16,000 households across 25 provinces, municipalities, and autonomous regions of China, representing up to 95% of 6–36 month-old Chinese. To calculate stunting prevalence, we standardized height by age and gender using the Z-score method: following the World Health Organization guidelines, we defined stunting as height-for-age Z score (HAZ) below minus 2SD [28]. CFPS also presents self-reported data on household income per capita, which we compared with the poverty line of ¥19.4 RMB per day [29]. We could then identify whether a child in a household lived in poverty or not. CFPS did not collect information on whether the children were already receiving YYB or not. However, only around 1% of children were covered by the program in 2015 [30], indicating very minimal impact of the existing program. Therefore, we assumed that the children included in the CFPS were not covered by the YYB program.

### 2.3. Estimation of Health Benefits

The health gains were defined as the number of child stunting cases that would be averted by YYB per province and wealth group: First, we estimated the population aged 6–36 months in province p and with poverty status k, denoted POP_p,k_. Second, for each child (denoted “i”) in the dataset, we checked his/her original height (Height_pre,p,k,i_) and standardized it with child age and sex to obtain HAZ. We determined whether the child was stunted by examining whether his/her HAZ (HAZ_pre,p,k,i_) was below minus 2SD. The stunting prevalence without hypothetical rollout of YYB in province p for poverty status k, denoted S_pre,p,k_, was the probability for a child in that subgroup to be stunted. Third, we used an estimate of YYB effectiveness [21] (denoted Eff) to estimate the impact of YYB on child height. We assumed Eff to be the same for all children. The likelihood for a child to be covered by YYB is denoted as Cov_p_ (for province p). The expected post-intervention height of child i (Height_post,p,k,i_) could be expressed using the following static model:Height_post,p,k,I_ = Height_pre,p,k,i_ + Eff * Cov_p_(1)

We checked the post-intervention stunting status of each child (HAZ_post,p,k,i_) by examining whether Height_post,p,k,i_ was below minus 2SD. We then estimated post-intervention stunting prevalence in province p and for poverty status k (S_post,p,k_) by counting how many children had HAZ_post,p,k,i_ below minus 2SD. Hence, the number of stunting cases averted (AVERT_p,k_) is calculated as:AVERT_p,k_ = POP_p,k_ * (S_pre,p,k_ − S_post,p,k_)(2)

### 2.4. Estimation of Costs

According to previous reports and studies [31,32,33], the YYB intervention would include four cost components: procurement and manufacturing costs; implementation costs; advertisement costs; and transportation costs (see Appendix A for further detail). The first three cost components were largely fixed, while transportation costs were assumed to vary based on transportation availability and accessibility across provinces and the type of YYB delivery to children and their households.

Procurement and manufacturing costs (M) were estimated at 35.0 cents (RMB; 9.9 cents ($)) per YYB pack, including manufacturing, printing for educational materials, personnel training for distribution and storage [31,32,33]. Implementation costs (F) included the costs imposed on township hospitals, village health posts and other organizations required to coordinate the delivery of YYB: F was estimated at 17.9 cents (RMB; 5.1 cents ($)) per pack [31]. Advertisement costs (A) were estimated at 16.7 cents (RMB; 4.7 cents ($)) per pack and covered raising public awareness and training of staff for implementation [31].

The transportation costs in province p and for poverty status k were denoted T_p,k_, which included the delivery costs from manufacturing companies to eligible households. T_p,k_ was composed of two parts (Figure 1). The first is the transportation costs from manufacturing companies to township hospitals (TM_p_), and these costs were borne by manufacturing companies. According to published studies [31,32,33], TM_p_ would vary greatly depending on accessibility and distance to township hospitals, from about 0.9 to 13.5 cents (RMB; 0.3 to 3.8 cents ($)) per YYB pack. Second, the costs from township hospitals to households, which could be materialized in two ways depending on the delivery type. There were two main ways to deliver YYB, denoted “Delivery Type 1” and “Delivery Type 2” [31]. Delivery Type 1 corresponds to the situation where village doctors come to the hospitals every month to collect YYB, and village health posts serve as the distribution location for caregivers to obtain YYB for their children. The costs borne by the village doctors (TV_p_ in province p) would be subsidized by the government [31]. We accounted for the time losses borne by the caregivers (TH_p,k_, for households of wealth status k in province p). Delivery Type 2 corresponds to the situation where caregivers come to township hospitals monthly to get YYB at their own expenses (TO_p,k_).

Transportation costs (T_p_) were expected to vary with two major factors: accessibility and distance from manufacturing companies to township hospitals and delivery type. For the first factor, road density was a critical determinant [31,34]. A number of studies on forest management used road density to estimate transportation costs [34,35,36,37,38]. We followed these studies [37,39] and modeled transportation costs from manufacturing companies to township hospitals in province p (TM_p_) based on the local road network density (denoted R_p_) (Appendix A). We assumed transportation costs would increase linearly with lower road density: the province with the largest R_p_ (Shanghai, 208 km per 100 km^2^ of land area) would face the lower transportation costs (0.9 cents (RMB; 0.3 cents ($)) per pack), while the province with the lowest R_p_ (Gansu, 31 km per 100 km^2^) would face the higher transportation costs (13.5 cents (RMB; 3.8 cents ($)) per pack). Subsequently, we could express the transportation costs from manufacturing companies to township hospitals (in cents, RMB) per pack in province p as: TM_p_ = 13.5 − (13.5 − 0.9)/(208 − 31)*(R_p_ − 31). For the second component, we modeled two delivery types. Delivery Type 1 is where village doctors would come to township hospitals every month to collect YYB at 0.9 cents per pack (RMB; 0.3 cents ($)) [31]. Caregivers would come to village health posts monthly to obtain YYB for their children. Following previous practices [2,40,41,42,43], we calculated caregivers’ time costs (TH_p,k_) in the following way: we multiplied caregivers’ travel time to health posts with provincial-level GDP per capita. Caregivers’ travel time was sourced from CFPS data and provincial-level GDP per capita data from the China Statistical Yearbook. Delivery Type 2 is where caregivers would come to township hospitals monthly to obtain YYB. The time costs to township hospitals (TO_p,k_) were borne by caregivers: CFPS did not collect information on travel time to township hospitals, so we imputed the travel time to the nearest town. We multiplied travel time by province-level GDP per capita to estimate the time costs for caregiver travel to township hospitals (TO_p,k_).

The estimated total transportation costs could then be expressed as: T_p,k_ = TM_p_ + TV_p_ + TH_p,k_ for Delivery Type 1 and T_p,k_ = TM_p_ + TO_p,k_ for Delivery Type 2. Subsequently, the total costs per pack could be calculated as: PP_p,k_ = M + F + A + T_p,k_. Lastly, the total costs per province p and wealth status k, for 12 months (365 days), could be estimated as PP_p,k_ multiplied by POP_p,k_, Cov_p,k_, 365 and the number of YYB packs needed for a 12-month intervention.

All types of costs were expressed both in 2011 RMB and international dollars. An incremental cost-effectiveness ratio was computed in terms of cost per stunting case averted.

### 2.5. Sensitivity Analyses

To test the robustness of our findings, we conducted two sets of sensitivity analyses. First, we examined the impact of YYB effectiveness by halving the original effectiveness Eff. This sensitivity analysis could capture variations in Eff potentially caused by household adherence and acceptability: previous studies showed that adherence of YYB program was around 80% and acceptability around 70% [8,9]. Likewise, we further tested a YYB effectiveness set to the lower bound of the 95% confidence interval of the estimated effect size from the meta-analysis (i.e., 0.96 cm) [21]. Second, we studied the impact of transportation costs by either halving or doubling transportation costs for Delivery Types 1 and 2, respectively. This sensitivity analysis could materialize economies of scale: transportation costs per capita may decrease with increasing coverage (e.g., from 25% to 75%); in addition, if many households lived in remote or mountainous areas, transportation costs per capita might increase with higher YYB coverage. In other words, this sensitivity analysis could test the variability in transportation costs across settings. For each sensitivity analysis, we reported on the resulting variations in costs, health benefits and cost-effectiveness in province p and for wealth group k.

## 3. Results

We first report on the different components of transportation costs (Appendix A). The average transportation costs from manufacturing companies to township hospitals were estimated to be 8.1 cents (RMB; 2.3 cents ($)) per pack. On average, caregivers’ transportation costs to health posts were 0.9 cents (RMB; 0.3 cents ($)) per pack, while caregivers’ transportation costs to township hospitals were estimated at 8.3 cents (RMB; 2.4 cents ($)) per pack. Appendix A present the estimated transportation costs by YYB delivery type (Type 1 vs. Type 2). In all provinces, the transportation costs were lower with Delivery Type 1 (9.9–10.2 cents (RMB; 2.8–2.9 cents ($)) on average) compared to Delivery Type 2 (16.8–25.2 cents (RMB; 4.8–7.2 cents ($)) on average). We estimated the costs per pack for each province and wealth group (Appendix A), as well as the total estimated costs at 25% and 75% coverage levels (Appendix A). At 25% coverage, the total costs would be estimated at ¥0.8–1.0 billion ($0.23–0.28 billion) for Delivery Type 1, compared with ¥0.9–1.2 billion ($0.26–0.34 billion) for Delivery Type 2.

Second, we estimated the health gains conferred by YYB Before YYB intervention, stunting prevalence would vary substantially by province and poverty status (Appendix A): Yunnan had the highest stunting prevalence at 27% (poverty group), while Beijing, Tianjin and Shanghai had the lowest stunting prevalence (about null). With 25% coverage of YYB, the estimated stunting prevalence in Yunnan would go down to 24% (poverty group); with 75% coverage, it would further decrease to 16% (Appendix A).

Overall, there was about 4.0 million stunted children before YYB intervention, including 2.7 million among the poverty group. Henan had the largest number of stunted children (about 0.4 million), while Beijing, Tianjin, Shanghai and Zhejiang had the lowest numbers (cumulatively less than 10,000) (Appendix A). The estimated impact of YYB would then vary greatly by province and wealth status. Nationally, with 25% coverage, around 0.6 million stunting cases could be averted, including 0.4 million cases for the poverty group; with 75% YYB. coverage, a total of 1.9 million stunting cases could be averted, including 1.3 million cases for the poverty group. Jiangxi would rank first in terms of stunting cases averted (about 80,000), while Beijing, Tianjin, Shanghai and Zhejiang would rank last (cumulatively less than 3000) (see Table 1).

We present the cost per stunting case averted per province and poverty group in Table 2 (and Appendix A). The cost per stunting case averted was generally lower with YYB Delivery Type 1. Among all provinces, targeting children in poverty in Chongqing (25% coverage, Type 1) would yield the lowest cost per stunting case averted (about ¥800 or $220 per stunted case). Conversely, targeting children not in poverty in Jilin province with Delivery Type 2 would yield the highest cost per stunting case averted (about ¥23,300 or $6600).

### Sensitivity Analyses

First, when halving the effect size of YYB, as expected, the cost per stunting case averted increased. The cost per stunting case averted was maintained lower with Delivery Type 1 (than with Delivery Type 2). The cost per stunting case averted remained lowest in Chongqing (about ¥1600 or $450 with Delivery Type 1 targeting children in poverty), while it would remain highest in Anhui (about ¥51,200 ($14,500) with Delivery Type 2 targeting children not in poverty) (Table 3 and Appendix A). Likewise, when we reduced the effect size of YYB to 0.96 cm (the lower bound of the estimated effect size from the meta-analysis [21]), the cost per stunting case averted increased, but with Chongqing and Anhui remaining the lowest and highest, respectively (Table 4 and Appendix A). Third, when the cost of YYB was halved, our key distributional findings would remain the same (Table 5 and Appendix A). Notably, targeting children in poverty in Chongqing with Delivery Type 1 would still yield the lowest cost per stunting case averted (about ¥750 ($210)). When the cost of YYB was doubled, our distributional findings would hold as well (Table 5 and Appendix A).

## 4. Discussion

We assessed the distributional consequences of the potential rollout of a YYB program on stunting prevalence in China. We found that the estimated cost-effectiveness would vary substantially by province, poverty status, delivery method and YYB coverage, from as low as ¥800 ($220) per stunting case averted (Chongqing) to as high as ¥23,000 ($6600, Jilin province). Chongqing, Jiangxi, Sichuan, Guangdong and Guangxi would generally present a lower cost to avert a stunting case (greater cost-effectiveness), while Anhui, Gansu and Jilin would present a higher cost (lower cost-effectiveness).

Moreover, in most provinces, the cost to avert a stunting case would be cheaper among children living in poverty, pointing to both the efficient and pro-poor potential of YYB rollout in China. Although the majority of the pilot YYB experiments recruited children only based on age (e.g., all children aged 6–36 months in selected counties) without taking household income into consideration [15,16,17,18,19,20,21], our results suggest that targeting the poor population might be potentially more cost-effective and equity-enhancing in China. However, we should note that many logistical issues would need to be considered when targeting the poor only, which might elevate the cost, such as identifying the poor and reaching out to them. Hence, for provinces where the poor are already identified and involved in other welfare programs, policymakers might consider only targeting the poor and adding YYB to the already existing welfare package.

We also found that the cost per stunting case averted would be lower in almost all provinces if YYBs were collected by village doctors from township hospitals and then caregivers would obtain them from village doctors, compared to caregivers coming to township hospitals to collect YYBs directly. Our study accounted for the costs borne by caregivers and found it more costly when caregivers traveled to township hospitals to obtain YYB (Delivery Type 2). Delivery Type 1 is the dominant delivery method in China [31]; however, Delivery Type 2 would lower government costs [44]. Therefore, policymakers should choose YYB coverage and delivery type based on both the government financial capacity and the willingness of caregivers to bear a share of the total cost.

Nevertheless, our analysis presents a number of limitations. First, there is uncertainty in several of our input parameters, including, for example transportation costs from manufacturing companies to township hospitals (TM). TM was based on the assumptions that it would increase linearly with lower road density: although similar modeling was applied previously [34,35,36,37,38], TM may not vary linearly with road density, but discontinuously as transport means used may change with road density (e.g., trucks could play a dominant role in places with higher road density) [45]. Although our sensitivity analyses allowed for variation in transportation costs, due to limited data, we still assumed similar TM costs for reaching all households within a province. Second, YYB coverage was assumed to be either 25% or 75%, and we neither sought an optimal coverage target (i.e., most cost-effective) for each province nor questioned the feasibility of our assumed coverage rates. Third, the economic evaluation we conducted is only one element toward priority setting, and decision-making is a complicated process. In addition to cost-effectiveness analysis or ECEA, decision-making processes should incorporate multiple considerations such as ethical, social and political factors. However, we assessed the likely equity impact of the YYB intervention, which is essential in a country with enormous heterogeneity such as China. Fourth, our study only considered rural China and thus ignored the country’s large urban–rural disparities. Moreover, we adopted the World Bank’s international poverty line of $5.50 per day, which could be revisited. For example, the GDP per capita in Jiangsu was around ¥95,000 ($26,000) in 2016, more than three times higher than in Gansu the same year (about ¥28,000 or $7000); hence, using the same poverty line across provinces may present some limitations. Fifth, although we considered adherence rates in our sensitivity analyses, we were uncertain how adherence would specifically affect effectiveness. For example, if children only took YYB for nine months (instead of 12), the adherence rate would be 75%, but the effectiveness might be maintained similar to the effectiveness of a 12-month intervention. Lastly, our study assumed that none of the children included in the CFPS survey were already covered by the YYB program, which might lead to an underestimation of the stunting cases that could be averted by YYB rollout, as well as an underestimation of the program effectiveness.

Despite the limitations, our study is novel in that it clearly stresses the distributional and equity implications of potential YYB rollout in China and points to the specific locations and provinces where YYB should be prioritized in terms of both efficient and equitable allocation. Our work highlights the pro-poor nature of YYB and how its rollout could redistribute population health in large and unequal low- and middle-income countries such as China. Future work should expand on our analysis by incorporating essential features capturing the feasibility and acceptability of different delivery scenarios. Furthermore, the effects of YYB can be multifaceted, including reducing anemia prevalence, strengthening anthropometric growth and improving cognitive development, which would add to the overall benefits of YYB beyond solely averting stunting. Future studies should be conducted to reveal the full broader impact of YYB

## Figures and Tables

**Figure 1 nutrients-12-02643-f001:**
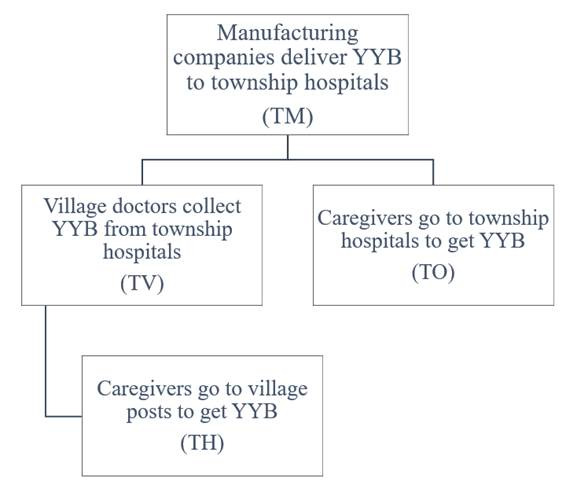
The types of delivery and associated costs for YYB, from manufacturing companies to eligible children.

**Table 1 nutrients-12-02643-t001:** Number of stunting cases averted by YYB program, by province and poverty status (25% and 75% coverage).

Province	25% Coverage		75% Coverage	
	Below Poverty Line	Above Poverty Line	Below Poverty Line	Above Poverty Line
Anhui	4074	5431	25,799	9310
Beijing	0	0	0	0
Chongqing	24,076	0	29,694	0
Fujian	4982	4095	18,286	25,516
Gansu	9424	1261	29,779	5374
Guangdong	32,623	20,678	76,022	57,978
Guangxi	33,537	5598	68,566	43,095
Guizhou	14,453	6509	54,751	22,093
Hebei	48,005	6549	121,902	20,565
Heilongjiang	8310	0	34,462	0
Henan	11,561	24,716	117,388	91,953
Hubei	5958	5240	30,902	37,334
Hunan	7273	16,428	71,374	55,224
Jiangsu	5722	6383	23,250	32,320
Jiangxi	33,834	45,538	100,104	76,668
Jilin	3392	746	14,379	13,234
Liaoning	4588	1224	14,681	4468
Shaanxi	11,207	2074	43,072	13,958
Shandong	33,804	4243	109,862	27,094
Shanghai	0	0	0	0
Shanxi	8710	4296	31,960	23,949
Sichuan	48,825	18,957	146,474	28,088
Tianjin	0	0	0	0
Yunnan	30,226	3318	106,351	33,686
Zhejiang	2955	0	6456	0
Total	387,538	183,284	1,275,515	621,907

**Table 2 nutrients-12-02643-t002:** Cost-effectiveness of YYB program, by province and poverty status (Cost (¥) per stunting case averted).

Province	25% Coverage				75% Coverage			
	Delivery Type 1		Delivery Type 2		Delivery Type 1		Delivery Type 2	
	Below Poverty Line	Above Poverty Line	Below Poverty Line	Above Poverty Line	Below Poverty Line	Above Poverty Line	Below Poverty Line	Above Poverty Line
Anhui	11,684	9943	18,385	13,184	5531	17,401	8710	23,083
Beijing	NA	NA	NA	NA	NA	NA	NA	NA
Chongqing	785	NA	1213	NA	1906	NA	2953	NA
Fujian	4175	5983	4476	6325	3418	2877	3653	3049
Gansu	3417	11,023	3661	10,864	3244	7741	3472	7648
Guangdong	1858	2331	2474	2486	2393	2494	3183	2658
Guangxi	2013	7556	2362	7931	2955	2945	3465	3093
Guizhou	3660	3595	4248	3564	2900	3177	3364	3155
Hebei	1733	7299	2075	7757	2048	6968	2450	7411
Heilongjiang	2756	NA	2888	NA	1991	NA	2086	NA
Henan	9558	3799	10,077	3993	2823	3065	2976	3219
Hubei	4851	8760	5421	10,267	2809	3688	3136	4323
Hunan	7851	3646	9762	4273	2401	3256	2983	3814
Jiangsu	2936	8773	3093	9243	2168	5195	2284	5473
Jiangxi	1578	968	1699	1058	1599	1724	1724	1886
Jilin	4186	18,901	4511	23,324	2963	3189	3199	3952
Liaoning	3684	11,846	5536	11,846	3447	9714	5184	9714
Shaanxi	2954	6895	3650	7859	2305	3074	2851	3496
Shandong	2000	13,128	2151	13,080	1846	6171	1985	6145
Shanghai	NA	NA	NA	NA	NA	NA	NA	NA
Shanxi	2859	4493	3042	4516	2340	2422	2491	2434
Sichuan	2153	1445	2245	1704	2153	2923	2245	3453
Tianjin	NA	NA	NA	NA	NA	NA	NA	NA
Yunnan	2309	6691	3487	7294	1969	1977	2973	2158
Zhejiang	2707	NA	2707	NA	3702	NA	3717	NA

Notes: Delivery Type 1: Manufacturing companies deliver YYB to township hospitals; then, village doctors come to township hospitals to collect YYB monthly and caregivers get YYB from village health posts. Delivery Type 2: Manufacturing companies deliver YYBs to township hospitals; then, caregivers come to township hospitals monthly to obtain YYB. NA means the values are not available.

**Table 3 nutrients-12-02643-t003:** Cost-effectiveness of YYB program, by province and poverty status, when the effect size of YYB is halved (Cost (¥) per stunting case averted).

Province	25% Coverage				75% Coverage			
	Delivery Type 1		Delivery Type 2		Delivery Type 1		Delivery Type 2	
	Below Poverty Line	Above Poverty Line	Below Poverty Line	Above Poverty Line	Below Poverty Line	Above Poverty Line	Below Poverty Line	Above Poverty Line
Anhui	18,421	38,571	28,997	51,179	11,052	24,302	17,397	32,233
Beijing	NA	NA	NA	NA	NA	NA	NA	NA
Chongqing	1593	NA	2468	NA	3987	NA	6179	NA
Fujian	6662	16,118	7127	17,107	6334	6059	6774	6428
Gansu	5823	44,268	6231	43,910	6907	15,083	7392	14,924
Guangdong	4371	5087	5815	5419	4538	4784	6038	5096
Guangxi	3788	11,137	4442	11,703	6181	6093	7249	6402
Guizhou	5851	9590	6786	9520	5662	6959	6567	6906
Hebei	4043	11,384	4838	12,110	4110	12,053	4918	12,823
Heilongjiang	5158	NA	5409	NA	3961	NA	4154	NA
Henan	19,423	9672	20,478	10,157	5407	6102	5701	6408
Hubei	11,120	15,866	12,414	18,597	5638	7405	6295	8678
Hunan	12,386	8699	15,391	10,190	5031	6353	6250	7442
Jiangsu	4777	10,067	5028	10,610	4607	10,145	4849	10,693
Jiangxi	2878	1995	3104	2181	3204	3447	3455	3769
Jilin	9073	37,802	9789	46,774	6100	5918	6582	7333
Liaoning	14,457	27,002	21,728	27,002	8501	18,796	12,779	18,796
Shaanxi	6167	14,314	7632	16,284	4468	6256	5529	7117
Shandong	3947	33,333	4246	33,194	3712	14,167	3992	14,113
Shanghai	NA	NA	NA	NA	NA	NA	NA	NA
Shanxi	5164	12,918	5498	12,977	4609	4699	4906	4722
Sichuan	4469	3074	4658	3632	4545	6024	4737	7117
Tianjin	NA	NA	NA	NA	NA	NA	NA	NA
Yunnan	4221	9083	6373	9926	3844	4140	5803	4524
Zhejiang	5559	NA	5587	NA	8093	NA	8133	NA

Notes: Delivery Type 1: Manufacturing companies deliver YYB to township hospitals; then, village doctors come to township hospitals to collect YYB monthly and caregivers get YYB from village health posts. Delivery Type 2: Manufacturing companies deliver YYB to township hospitals; then, caregivers come to township hospitals monthly to obtain YYB.

**Table 4 nutrients-12-02643-t004:** Cost-effectiveness of YYB program, by province and poverty status, when the effect size of YYB is reduced to 0.96 cm (Cost (¥) per stunting case averted).

Province	25% Coverage				75% Coverage			
	Delivery Type 1		Delivery Type 2		Delivery Type 1		Delivery Type 2	
	Below Poverty Line	Above Poverty Line	Below Poverty Line	Above Poverty Line	Below Poverty Line	Above Poverty Line	Below Poverty Line	Above Poverty Line
Anhui	20,877	55,670	33,907	61,092	12,122	28,551	21,457	35,357
Beijing	NA	NA	NA	NA	NA	NA	NA	NA
Chongqing	1934	NA	2605	NA	4800	NA	6625	NA
Fujian	8251	19,679	8001	17,899	7141	6816	7041	7759
Gansu	6189	46,179	7453	56,612	9198	17,620	9356	19,590
Guangdong	5391	5402	8015	5923	5255	5632	6926	5614
Guangxi	4328	12,646	5415	12,444	6584	6984	8542	7652
Guizhou	6921	10,919	7675	11,588	6763	7799	8439	8393
Hebei	4890	13,634	5408	13,478	5459	13,784	5565	14,377
Heilongjiang	6659	NA	6879	NA	4156	NA	4569	NA
Henan	24,539	12,313	27,239	10,666	5687	7362	6532	6844
Hubei	11,446	16,776	13,168	18,805	6759	8017	8210	10,621
Hunan	12,913	9917	17,688	12,988	6809	8060	7738	8749
Jiangsu	5411	10,600	5607	11,223	5882	10,400	5976	13,255
Jiangxi	3099	2438	3587	2606	3947	3635	4428	4323
Jilin	10,931	46,689	10,275	60,000	6804	7082	6745	8341
Liaoning	21,338	29,234	26,625	34,198	10,775	23,012	14,468	24,178
Shaanxi	8187	18,263	7792	17,698	5260	7416	6918	9204
Shandong	4603	41,443	5430	42,857	4682	19,065	5245	19,242
Shanghai	NA	NA	NA	NA	NA	NA	NA	NA
Shanxi	5950	18,363	6996	15,109	5688	5533	5841	5661
Sichuan	5798	3575	5784	3960	5123	6915	5921	8785
Tianjin	NA	NA	NA	NA	NA	NA	NA	NA
Yunnan	5455	10,302	7399	10,867	4113	5167	6899	5454
Zhejiang	7435	NA	6375	NA	8958	NA	9566	NA

Notes: Delivery Type 1: Manufacturing companies deliver YYB to township hospitals; then, village doctors come to township hospitals to collect YYB monthly and caregivers get YYB from village health posts. Delivery Type 2: Manufacturing companies deliver YYB to township hospitals; then, caregivers come to township hospitals monthly to obtain YYB.

**Table 5 nutrients-12-02643-t005:** Cost-effectiveness of YYB program, by province and poverty status, when the transportation costs (for both Delivery Type 1 and Delivery Type 2) are halved or doubled (Cost (¥) per stunting case averted).

Province	25% Coverage				75% Coverage			
	Delivery Type 1		Delivery Type 2		Delivery Type 1		Delivery Type 2	
	Below Poverty Line	Above Poverty Line	Below Poverty Line	Above Poverty Line	Below Poverty line	Above Poverty line	Below Poverty line	Above Poverty line
	The Cost per Stunting Case Averted Assuming the Transportation Costs of YYB Were Halved (top)/Doubled (bottom)							
Anhui	11,133	9510	14,486	11,133	5273	16,642	6862	19,482
Beijing	NA	NA	NA	NA	NA	NA	NA	NA
Chongqing	745	NA	960	NA	1811	NA	2336	NA
Fujian	3894	5568	4039	5749	3182	2681	3301	2768
Gansu	3120	9967	3240	9909	2962	7014	3076	6973
Guangdong	1754	2205	2061	2281	2258	2359	2654	2440
Guangxi	1830	6875	2004	7067	2685	2679	2940	2754
Guizhou	3422	3384	3715	3371	2710	2991	2942	2979
Hebei	1622	6821	1792	7054	1916	6517	2117	6739
Heilongjiang	2513	NA	2581	NA	1818	NA	1867	NA
Henan	9177	3646	9436	3741	2711	2940	2788	3017
Hubei	4544	8349	4827	9102	2628	3516	2792	3833
Hunan	7379	3389	8332	3702	2256	3024	2547	3303
Jiangsu	2815	8403	2892	8640	2078	4979	2135	5119
Jiangxi	1471	903	1533	948	1492	1610	1555	1690
Jilin	3838	17,372	4003	19,630	2716	2936	2834	3318
Liaoning	3417	11,005	4342	11,005	3203	9046	4071	9046
Shaanxi	2747	6413	3097	6886	2144	2859	2417	3070
Shandong	1907	12,464	1983	12,439	1760	5856	1830	5844
Shanghai	NA	NA	NA	NA	NA	NA	NA	NA
Shanxi	2668	4196	2760	4207	2181	2258	2257	2264
Sichuan	1984	1334	2030	1465	1984	2700	2030	2965
Tianjin	NA	NA	NA	NA	NA	NA	NA	NA
Yunnan	2135	6180	2723	6489	1820	1826	2322	1917
Zhejiang	2525	NA	2531	NA	3466	NA	3475	NA
Anhui	12,775	10,820	26,189	17,311	6051	18,934	12,405	30,295
Beijing	NA	NA	NA	NA	NA	NA	NA	NA
Chongqing	862	NA	1724	NA	2096	NA	4193	NA
Fujian	4749	6781	5330	7508	3882	3265	4356	3615
Gansu	4009	13,032	4488	12,801	3806	9171	4261	9008
Guangdong	2067	2585	3295	2890	2661	2766	4242	3092
Guangxi	2380	8918	3076	9683	3492	3475	4513	3773
Guizhou	4143	4014	5313	3959	3281	3548	4208	3499
Hebei	1957	8231	2638	9162	2312	7864	3117	8753
Heilongjiang	3225	NA	3493	NA	2333	NA	2527	NA
Henan	10,315	4112	11,351	4493	3048	3316	3354	3623
Hubei	5475	9581	6606	12,593	3167	4034	3821	5302
Hunan	8808	4165	12,616	5415	2692	3717	3857	4833
Jiangsu	3186	9495	3495	10,442	2352	5626	2581	6187
Jiangxi	1789	1097	2036	1277	1814	1954	2065	2276
Jilin	4883	21,854	5545	30,888	3456	3694	3925	5221
Liaoning	4199	13,422	7900	13,422	3937	11,033	7406	11,033
Shaanxi	3365	7846	4767	9739	2627	3498	3721	4342
Shandong	2185	14,472	2487	14,371	2017	6799	2296	6751
Shanghai	NA	NA	NA	NA	NA	NA	NA	NA
Shanxi	3249	5110	3618	5155	2656	2750	2958	2774
Sichuan	2492	1663	2675	2187	2492	3368	2675	4428
Tianjin	NA	NA	NA	NA	NA	NA	NA	NA
Yunnan	2659	7707	5013	8946	2267	2277	4274	2643
Zhejiang	3042	NA	3069	NA	4176	NA	4213	NA

Notes: Delivery Type 1: Manufacturing companies deliver YYB to township hospitals; then, village doctors come to township hospitals to collect YYB monthly and caregivers get YYB from village health posts. Delivery Type 2: Manufacturing companies deliver YYB to township hospitals; then, caregivers come to township hospitals monthly to obtain YYB.

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
