# Peer review of "Equity and Distributional Impact on Stunting of a Nutritional Package Targeting Children Aged 6–36 Months in China: Findings from a Modeling Study"

_nutrients, 2020, doi:10.3390/nu12092643_

Round 1
Reviewer 1 Report
This manuscript does an excellent job presenting relevant data on nutritional package costs and highlighting its health benefits in vulnerable Chinese population. Authors add novelty to this topic by evaluating a model of distribution to improve cost-effectiveness and guarantee its availability.It is well written, only minor observations to consider.
- Page 2: Manuscript title is not consistent with cover page title.
- Please add CFPS: China’s Family Panel Studies to Abbreviations.
- Please substitute yingyangbao for Ying Yang Bao throughout the manuscript.
- Consider adding dollar currency next to yen currency for consistency of the manuscript.
- Page 3, Line 86: Please consider substituting “in under-five children” to: in children under age 5.
- Page 9, Line 219: Typo error, change 209 for 208.
- It would be interesting if authors could provide more information related to the method used by caregivers to prepare nutritional package. Also, if there are any other criteria besides being in the age group to be eligible to receive the nutritional package.
- Notes under table 1 should be deleted because they do not belong to that table. Most likely a formatting error.
- Tables in pages 20 and 21 are both labeled as Table 5.
- Please verify that the information in the cover page related to tables corresponds with what is found in the supplementary material (only 6 tables were attached).
Author Response
This manuscript does an excellent job presenting relevant data on nutritional package costs and highlighting its health benefits in vulnerable Chinese population. Authors add novelty to this topic by evaluating a model of distribution to improve cost-effectiveness and guarantee its availability. It is well written, only minor observations to consider.
- Response: We thank the reviewer for her/his positive comments on our manuscript.
Page 2: Manuscript title is not consistent with cover page title.
- Response: We have revised the manuscript title to be “Equity and distributional impact on stuntingof a nutritional package targeting children aged 6-36 months in China: findings from a modeling study”
Please add CFPS: China’s Family Panel Studies to Abbreviations.
- Response: We have added “China Family Panel Studies” to the Abbreviation list.
Please substitute yingyangbao for Ying Yang Bao throughout the manuscript.
- Response: We have substituted Yingyangbao for Ying Yang Bao throughout the manuscript.
Consider adding dollar currency next to yen currency for consistency of the manuscript.
- Response: We have added $ currency to all the RMB currency numbers. For the tables using RMB (i.e. Tables 2-5), we have converted all the figures to 2011 Purchasing Power Parity $ and we present the corresponding results in Appendix Tables A12-A15.
Page 3, Line 86: Please consider substituting “in under-five children” to: in children under age 5.
- Response: We have replaced “in under-five children” with “in children under age 5”.
Page 9, Line 219: Typo error, change 209 for 208.
- Response: Thanks! We have replaced “209” with “208”.
It would be interesting if authors could provide more information related to the method used by caregivers to prepare nutritional package. Also, if there are any other criteria besides being in the age group to be eligible to receive the nutritional package.
- Response: We have now added the method to be used by caregivers to prepare YYB in the Introduction section:
“YYB can be directly added to children’s meals (e.g., mushy food, soup, noodle), or reconstituted with water.10”
For almost all current YYB programs/pilots, children were selected by age without the use of other additional criteria.
Notes under table 1 should be deleted because they do not belong to that table. Most likely a formatting error.
- Response: Thank you. We have now deleted the note under table 1.
Tables in pages 20 and 21 are both labeled as Table 5.
- Response: This is the same table: the first part is labeled table 5 and the second part is labeled table 5 (continued). We have now highlighted the word “continued” in the table caption.
Please verify that the information in the cover page related to tables corresponds with what is found in the supplementary material (only 6 tables were attached).
- Response: We have verified that the supplementary materials and the list in the cover page are consistent.

Reviewer 2 Report
I commend the authors on a useful interesting peice of work relating to improving the health/growth of undernorusied children in China. There are also numerous details which, in my view, need correcting/clarifying and I've listed these below. I also recommend a thorough proof reading of the manuscript to iron out a few inconsistencies in grammar/syntax.
Detail
Line 78/79 revise wording: 'China is still one
of the countries with large numbers of stunted children' suggest: 'China has large numbers of stunted children'
Line 81: change the word 'equaling' to equal
Line 97: this sentence does not make sense: 'Compared to micronutrient and other complementary feeding supplements, such as sprinkles and crushable tablets, YYB is composed of soybean powder that provides both calories...' if you are comparing YYb with other potential altenatives tell the reader what is different? i.e. higher or lower in microutrients/energy which is what I think you intended?
Line 117: I think you should present figures in both Yen and dollar thorughout the manuscript, to help people contextualise the info- at the moment you mix the two.
Line 162: do you mean equal to or less than i.e: ⩽ ?
Line 183: see my point above Use both ¥ and $ throughout.
Line 192: I think you mean composed rather than decomposed...
Line 242: does this mean an exchange rate from 2011?!
Author Response
I commend the authors on a useful interesting peice of work relating to improving the health/growth of undernorusied children in China. There are also numerous details which, in my view, need correcting/clarifying and I've listed these below. I also recommend a thorough proof reading of the manuscript to iron out a few inconsistencies in grammar/syntax.
- Response: We thank the reviewer for her/his positive comments on our manuscript. We have thoroughly checked the manuscript to ensure consistency in grammar and syntax.
Line 78/79 revise wording: 'China is still one of the countries with large numbers of stunted children' suggest: 'China has large numbers of stunted children'
- Response: We have revised it to “China has large numbers of stunted children”.
Line 81: change the word 'equaling' to equal
- Response: We have revised “equaling” to “equal”.
Line 97: this sentence does not make sense: 'Compared to micronutrient and other complementary feeding supplements, such as sprinkles and crushable tablets, YYB is composed of soybean powder that provides both calories...' if you are comparing YYb with other potential altenatives tell the reader what is different? i.e. higher or lower in microutrients/energy which is what I think you intended?
- Response: We have revised this sentence accordingly:
“Compared to other complementary feeding supplements (e.g. sprinkles and crushable tablets) that only provide micronutrients, YYB contains soybean powder that provides both calories (usually ~50Kcal per pack) and protein (3g per pack) in addition to various micronutrients.7–9”
Line 117: I think you should present figures in both Yen and dollar thorughout the manuscript, to help people contextualise the info- at the moment you mix the two.
- Response: We have added $ currency to all the RMB currency numbers. For the tables using RMB (i.e. Tables 2-5), we have converted all the figures to 2011 Purchasing Power Parity $ and we present the corresponding results in Appendix Tables A12-A15.
Line 162: do you mean equal to or less than i.e: ⩽?
- Response: We have removed the symbol and used “below” instead.
Line 183: see my point above Use both ¥ and $ throughout.
- Response: See response above.
Line 192: I think you mean composed rather than decomposed...
- Response: Thank you for pointing this out. We have replaced “decomposed” by “composed”.
Line 242: does this mean an exchange rate from 2011?!
- Response: We have used 2011 RMB and 2011 Purchasing Power Parity $. We have now clarified this.
